# UV-B Radiation Effects on the Alpine Plant *Kobresia humilis* in a Qinghai-Tibet Alpine Meadow

**DOI:** 10.3390/plants11223102

**Published:** 2022-11-15

**Authors:** Shengbo Shi, Rui Shi, Tiancai Li, Dangwei Zhou

**Affiliations:** 1Key Laboratory of Adaptation and Evolution of Plateau Biology, Qinghai Provincial Key Laboratory of Restoration Ecology of Cold Area and Crop Molecular Breeding, Northwest Institute of Plateau Biology, Chinese Academy of Sciences, Xining 810001, China; 2State Key Laboratory Breeding Base of Desertification and Aeolian Sand Disaster Combating, Gansu Desert Control Research Institute, Lanzhou 730070, China; 3Guangdong Berkgen Biopharmaceuticals Co., Ltd., Shaoguan Advanced Institute of Biopharmaceuticals, Guangzhou 512000, China

**Keywords:** *Kobresia humilis*, leaf morphology, ozone depletion, photosynthesis, Qinghai-Tibet Plateau, UV-B radiation

## Abstract

Enhanced UV-B radiation resulting from stratospheric ozone depletion has been documented both globally and on the Qinghai-Tibet Plateau in China. The response of *Kobresia humilis*, an important alpine meadow plant species, to enhanced UV-B radiation was experimentally investigated at the Haibei Alpine Meadow Ecosystem Research Station (37°29′–37°45′ N, 101°12′–101°23′ E; alt. 3200 m). *K. humilis* was exposed to UV-B radiation including ambient UV-B and enhanced UV-B (simulating a 14% reduction in the ozone layer) in a randomized design with three replications of each treatment. Enhanced UV-B radiation resulted in a significant increase of both leaf area and fresh weight chlorophyll and carotenoid but had no effect on UV-B absorbing pigments. Similarly, enhanced UV-B radiation did not significantly change the photosynthetic O_2_ elevation rate while leaf thickness, width, and length significantly increased (*p* < 0.01). The enhanced UV-B radiation was associated with 2–3 days earlier flowering and a larger number of flowers per spikelet. The enhanced UV-B generally resulted in larger leaves and more flowers but earlier phenology. In summary, these findings suggest that alpine species of *K. humilis* have adapted to the strong solar UV-B radiation intensity presented on the Qinghai-Tibet Plateau, but the interspecies differences and their influence on trophic level should be more concerning.

## 1. Introduction

Stratospheric ozone depletion, caused by releases of man-made chemicals [1,2], increases incident UV-B radiation [3], which has been generally thought to have a negative impact on vegetation in both agricultural and natural ecosystems [4]. UV-B represents a small fraction of the energy but itself is highly energetic [5]. While the release of ozone depleting chemicals has mostly slowed or stopped, little resulting change has been detected in UV-B transmission [6]. The responses of vegetation to UV-B mainly include morphological and physiological changes associated with protective responses such as thicker leaves and increased protective pigments [7,8,9,10]. While conceptually UV-B has an overall negative effect, the observed effects have been typically subtle and spanned the whole range of responses: negative, neutral, and positive [5,11,12,13]. This implies that the overall UV-B effect will depend on the particular ecosystem [14,15]. The likelihood and magnitude of UV-B impacts will increase in areas that have both high ozone depletion and high environmental vulnerability [16,17]. 

The Qinghai-Tibet Plateau in China satisfies both criteria. First, regional ozone depletion has been documented [18] and the high elevation allows naturally higher amounts of UV-B [19,20]. Second, the Qinghai-Tibet Plateau has been known to be highly sensitive to environmental change due to its harsh alpine conditions [21] and the components of the ecosystem generally are near the edge of their tolerance [22]. Within the plateau, alpine meadows form a common and widely distributed plant community type [23,24]. *Kobresia humilis*, a perennial sedge (Cyperaceae) that is 6–15 cm tall with a crowded growth form [25], dominates these alpine meadows. *K. humilis*, due to its high palatability, supports a wide range of wild herbivores and is a primary forage plant for sheep and yak [23]. 

The environmental conditions in the Qinghai-Tibet Plateau region are harsh, with “high” and “cold” as its prominent natural features. Due to the height of altitude and thin air, unique climatic characteristics are formed, and the climate changes drastically: the wind is strong, the difference in annual temperature is small, but the difference in daily temperature is large, and solar radiation is strong, particularly the UV-B component. The extreme local conditions, similar to other studies such as Björn et al. [26] and Caldwell et al. [15], seemed likely to increase the negative effects of increased solar UV-B radiation. We hypothesized that native alpine species such as *K. humilis* would also be negatively impacted by an increase in UV-B radiation, including a reduction in their biomass or life span. To project the changes through the life cycle of this species, we focused on the season-long UV-B radiation exposure effects on the influence of photosynthesis, photosynthetic pigments, photomorphology, UV-B absorbing pigments, flowering phase, and leaf traits. 

## 2. Materials and Methods

### 2.1. Field Sites and Growth Conditions

The field experiment was conducted at the Haibei Alpine Meadow Ecosystem Research Station (37°29′–37°45′ N, 101°12′–101°23′ E). The study site is located at 3200 m asl, where the average barometric pressure is 691.4 hPa, the annual average air temperature is −1.7 °C (with the warmest month averaging 9.8 °C and the coldest month −14.8 °C), and the annual precipitation averages about 600 mm, with 80% falling during the growing season from May to September [23]. The shallow alpine meadow soil had an average thickness of 0.65 m. The surface 0.05–0.10 m horizons, which were Mat Crygelic Cambisols [27], were wet and high in organic matter.

The alpine *K. humilis* meadow is composed of perennial herbs; *K. humilis* is its constructive and dominant species. Main accompanying species include *Festuca ovina*, *Stipa grandis*, *Aster alpine*, *Gentianopsis paludosa*, *Saussurea superba*, *Thalictrum alpinum*, *Gueldenstaedtis diversifolia*, *Ligularia virgaurea*, etc. There was no grazing of livestock during the plant’s growing season from mid-May to late September.

### 2.2. Experimental Design and Irradiation

The UV-B study site was established in an alpine *Kobresia* meadow region near the research station. The methods generally follow Shi et al. [28]. Six metal frames (2.5 m × 1.3 m × 0.75 m in height) were arranged in a randomized design, with 3 replications of the enhanced UV-B treatment and 3 replications of ambient UV-B treatment. Each frame had six fluorescent lamps (UV-B-313, 40 W; Beijing Electric and Light Source Institute, Beijing, China). Enhanced UV-B radiation was achieved by wrapping 0.13 mm cellulose diacetate film (Courtauld Specialty Plastics, Derby, UK.) to block ecologically irrelevant UV-C (<280 nm). The cellulose diacetate film was replaced weekly. Ambient UV-B treatment consisted of unenergized lamps, which provided the same degree of shading as beneath treatment arrays. 

There were two field experiments that used the same methods. The first ran from 1999 to 2002. The second, which ran from 2004 to 2005, was located adjacent to the first experiment on previously untreated vegetation. UV-B radiation began in May and ended in September each year to match the beginning and end of the growing season. Lamps were illuminated for 7 h per day centered on local solar noon (about 13:15 Beijing time), and the enhanced UV-B_BE_ density was 15.80 kJ m^−2^ day^−1^ [28]. The enhanced UV-B treatment included ambient solar UV-B with the addition of the UV-B radiation provided by the fluorescent lamps. The UV-B treatment mimicked the daily increase in UV-B radiation likely to result from 14% ozone depletion at the study site based on the model described by Björn and Teramura [29]. 

### 2.3. Measurement

Morphological parameters including plant height, leaf thickness, leaf length, and leaf width were measured from healthy, mature leaves. Flower numbers and phenological phases were measured by dividing the 0.25 m^2^ square sampling area into 5 × 5 cm^2^ plots, and each *K. humilis* plant was marked within each small plot. The relative height, relative cover, relative density, and relative frequency were determined, and the Importance Value (IV) was calculated from the equation: IV = (height + coverage + density + frequency)/4.

At the end of September, all vegetation within the 0.25 m^2^ square was cut at soil level and sorted into individual species. After harvest and sorting, vegetation was dried to constant weight in an oven at 80 °C and the total above-ground biomass of *K. humilis* was measured. 

During plants vigorous growing stage, the photosynthetic O_2_ evolution rate was measured with an oxygen electrode (SP-2, Shanghai Plant Physiology Institute, Chinese Academy of Sciences). The middle fragments from different leaves were used for measuring the net photosynthetic O_2_ evolution rate using the methods of Li et al. [30]. The measurement conditions were: the temperature set to 25 ℃; the PAR in the chamber was 1800 μmol m^−2^ s^−1^ and supplied by a 150 W halogen lamp; and the CO_2_ in the solution was supplied by 25 mmol L^−1^ NaHCO_3_, which was dissolved in 50 mmol L^−1^ pH 7.5 phosphate buffer. 

Healthy, mature leaves were used for determining photosynthetic pigments and UV-B absorbing pigments. The samples from the middle of the leaves were measured both for leaf fresh weight and leaf area unit, respectively. To determine the photosynthetic pigments, samples were put into 10 mL of ethanol, acetone, and H_2_O (4.5:4.5:1 by volume) and stored in a darkened room with an average air temperature near 10 °C for 21 days [28]. The UV-B absorbing pigments were determined as the method described by Day et al. [31]. Samples were put into bottles containing 10 mL of methanol, HCl, and H_2_O (79:20:1 by volume) and stored in a darkened room as above for 21 days. Each measurement was repeated four times. Estimates of chlorophylls and carotenoids were made by measuring supernatant absorbance at 663, 645, and 440 nm with calculations according to methods of Wellburn [32]. A UV/visible spectrophotometer (UV-1601, Shimadzu) was used to estimate UV-B absorbing pigments in the 250–400 nm range. Since there were no qualitative changes in the pattern of spectral absorbance, the absorbance at 300 nm (*A*_300_) was chosen as the mean wavelength of the UV-B region and used for analysis.

### 2.4. Statistical Analysis

The independent-samples *t*-test was used to analyze differences between treatments. For photosynthetic O_2_ elevation rate, photosynthetic pigments, and UV-B absorbing pigments tests, the samples from all ambient UV-B and enhanced UV-B treatments were pooled. For photosynthetic O_2_ evolution rate, *n* = 5 and 6; and for other parameters, *n* = 4. All analyses were performed using SPSS (version 10.0, Chicago, IL, USA).

## 3. Results

After two and a half months of treatment with enhanced UV-B radiation, the leaf length and leaf width of the radiated plants were increased by 13.09% and 16.67% compared to that of the ambient UV-B level control plants, respectively (Table 1). The leaf thickness was also increased by 20% in the UV-B treated plant (*p* < 0.01). However, the plant height showed no significant difference between enhanced UV-B treated plants and those of ambient UV-B controlled *K. humilis* (Table 1). 

Enhanced UV-B radiation did not affect the Importance Values (IV) of *K. humilis* (Table 2). However, its values decreased from 4.58 to 3.98 after a second-growth seasonal exposure to enhanced UV-B. With a long-term increase in UV-B, the above-ground biomass of *K. humilis* showed a similar trend and decreased from 4.14 to 3.24 g per 0.25 m^2^.

The photosynthetic O_2_ evolution rate did not show a significant difference between the control and enhanced UV-B radiation treatments (Table 3). In terms of both the *K. humilis* leaf fresh weight and leaf area, the photosynthetic pigment contents of chlorophyll and carotenoids were significantly increased after exposure to enhanced UV-B radiation (*p* < 0.05). On a leaf fresh weight basis, enhanced UV-B radiation caused a 35.75% increase in chlorophylls and a 20.68% increase in carotenoids, while on a leaf area basis, there was a 31.51% increase in chlorophylls and a 17.04% increase in carotenoids. Although there was no statistical difference in UV absorbance, an increased tendency in the amount of UV-B absorbing pigments, whether based on the leaf area or fresh weight, was observed after enhanced UV-B radiation (a 12.60% and 42.48% increase, respectively). 

Supplemental UV-B radiation had no effect on the number of tillers between ambient UV-B and enhanced UV-B treatment (Table 4). Although the number of spikelet stems and the number of flowers had a significant increase as compared to ambient UV-B treatment (*p* < 0.05), the ratio of flowers to spikelet stem and the ratio of nutlets to spikelet stem only appeared to have an increasing tendency when compared with ambient UV-B control. Continuous observations between 16 May and 23 June 2004 indicated that enhanced UV-B radiation could promote flowering 2–3 days earlier; interestingly, the life span of flowering was not affected, because first flowering and fruiting times were both advanced similarly (about 2–3 days). 

## 4. Discussion

While many species (perhaps half of all species) have been shown to have a negative effect from enhanced UV-B radiation, some species appear unimpacted by additional UV-B radiation [15,17,33,34]. Possible mechanisms of UV-B adaptation include changes in plant structure or architecture; changes in the anatomy of the epidermal layer and epicuticular waxes; thicker and heavier leaves; and increases in pigments that absorb UV-B [4]. 

Our results confirmed that *K. humilis* is a UV-B resistant species [35] because the measured traits either were unaffected or improved under increased levels of UV-B radiation. Further, the photosynthetic O_2_ evolution rate of the leaves in *K. humilis* was not negatively influenced by exposure to enhanced UV-B radiation and, in some other repeated measurements, an increased photosynthetic O_2_ evolution rate was detected. The increased rate could be a consequence of the chemical and anatomical alteration of leaves. The UV-B resistance may also be attributed to the adaptation of alpine plants to the strong solar radiation intensity and the other harsh environmental factors present on the Qinghai-Tibet Plateau [28]. 

Our chlorophyll pigment results were consistent with the findings of Correia et al. [11], who found that enhanced UV-B radiation stimulated higher chlorophyll contents. This contrasts with a number of studies that found lower chlorophyll content with enhanced UV-B radiation, such as Tevini et al. [33]. Because inhibition of chlorophyll biosynthesis has not been associated with UV-B radiation [36,37], our increased chlorophyll content suggests a lack of photo-damage induced by UV-B radiation. This lack of damage may have been from leaf morphological traits such as the thicker epicuticular structure found by Zhou [38]. 

In the first 20 days after enhanced UV-B radiation, *K. humilis* showed no difference in plant height, leaf length, or leaf width compared to ambient UV-B; after 45 days, both leaf length and leaf width had an increasing trend in the enhanced UV-B treatment. While the day-by-day results are not presented (only the final measurements were shown in the tables), there was an increasing trend in plant height with enhanced UV-B radiation, especially near the end of the measurement period. Plant height has been shown to be important in mixed canopies where it represents competition for light [39]. Our increased leaf thickness in the UV-B exposed plants in comparison with the ambient UV-B treatment was similar to that of a wide range of species [40,41]. Plants appear to reduce the penetration of UV-B in the leaf by developing thicker leaves [31,40]. The increased leaf thickness may be important in protecting palisade layers from shortwave radiation and can also be accompanied by the adaxial epidermis of the leaves having enhanced UV-B-absorbing pigments [42]. These UV-B-absorbing pigments have been shown to be mainly flavonoids [34]. Our study showed that enhanced UV-B radiation increased *K. humilis* leaf length, leaf width, and potentially plant height but that these changes were accompanied by a 3–4 day reduction in the leaf lifespan. The enhanced UV-B may have given a competitive growth advantage, but at the cost of earlier leaf and plant senescence. The ecosystem level impact of these changes remains unclear because the cumulative morphological and phenological changes will change the interactions among plants and animals throughout the system [43].

At the study site, *K. humilis* flowers in the early spring, beginning in early May and lasting through mid-June. Our results showed that enhanced UV-B radiation not only promoted earlier flowering times but also increased the number of flowers per spikelet stem. Earlier flowering may not be important because *K. humilis* mainly relies on reproduction through vegetative propagation, but it does support the idea that enhanced UV-B changes the phenology across the whole growing season instead of only promoting earlier senescence [15]. While sexual reproduction appears rare for this species, future research in seed production and natural history under enhanced UV-B is needed to determine the long-term population impacts. 

Our results for *K. humilis* support the idea that species originating from regions with high solar radiation and harsh growing conditions exhibit protective or even positive responses to enhanced UV-B intensity compared to species originating in regions with milder growing conditions [20]. Our single species result does not indicate a negative effect of increasing UV-B radiation but, before UV-B changes to vegetation can be evaluated, more research is needed on the interspecies differences within the alpine *Kobresia* meadows with a concentration on impacts to community structure/function as well as biodiversity.

## Figures and Tables

**Table 1 plants-11-03102-t001:** Effects of enhanced UV-B on plant growth parameters of *K. humilis*.

Parameters		Ambient UV-B	Enhanced UV-B	*p*-Value	Sample Size
Plant height	cm	3.41 ± 0.58	3.49 ± 0.78	0.64	60
Leaf length	cm	2.98 ± 0.91	3.37 ± 0.71	0.01	60
Leaf width	cm	0.18 ± 0.06	0.21 ± 0.05	0.01	60
Leaf thickness	mm	0.20 ± 0.02	0.24 ± 0.05	0	120

Leaf thickness was measured on 15 August 2004; plant height, leaf length, and leaf width were measured in the middle of July 2004. Data represents effects from treatment beginning in early May 2004.

**Table 2 plants-11-03102-t002:** Effects of enhanced UV-B on the Importance Value and biomass of *K. humilis*.

Parameters		Ambient UV-B	Enhanced UV-B	Change (%)
IV		4.58	3.98	−13.10
Biomass	g	4.14	3.24	−21.74
	%	6.43	5.83	−9.33

The values were the means of three replicates. The above-ground biomass of *K. humilis* is per 0.25 m^2^. The data are from September 2005, which represents effects from two growing seasons of treatment.

**Table 3 plants-11-03102-t003:** Effects of enhanced UV-B on photosynthetic O_2_ evolution rate and pigments of *K. humilis*.

Parameters		Ambient UV-B	Enhanced UV-B	*p*-Value
*P* _n_	μmol O_2_ m^−2^ s^−1^	5.23 ± 1.14	6.44 ± 1.29	0.21
Chlorophyll	mg Chl g^−1^ (FW)	1.93 ± 0.09	2.62 ± 0.35	0.01
	μg Chl cm^−2^ (leaf area)	29.86 ± 1.54	39.27 ± 6.56	0.03
Carotenoids	mg Chl g^−1^ (FW)	0.58 ± 0.05	0.70 ± 0.06	0.02
	μg Chl cm^−2^ (leaf area)	9.04 ± 0.42	10.58 ± 1.68	0.05
UV-B absorbing pigments	A_300_ g^−1^ (FW)	4.92 ± 0.65	5.54 ± 1.32	0.44
	A_300_ cm^−2^/(leaf area)	1.13 ± 0.70	1.61 ± 0.29	0.25

The data were collected in the middle of July 2005, after the treatments had been in place for one year. For photosynthetic O_2_ evolution rate, *n* = 5 and 6; and for other parameters, *n* = 4. UV-B absorbing pigments were extracted with 79% acidic methane, measured at 50 mL/0.05 g and 10 mL/cm^2^.

**Table 4 plants-11-03102-t004:** Effects of enhanced UV-B on the number of tillers and flowers of *K. humilis*.

Parameters	Ambient UV-B	Enhanced UV-B	*p*-Value	Measured Time
Tiller number	540 ± 144	480 ± 100	0.587	1 July 2005
Number of spikelet stems	46.00 ± 12.52	73.33 ± 23.94	0.038	23 June 2004
Flower number	22.67 ±9.33	45.83 ± 7.59	0.021	4 June 2004
Flowers/spikelet stem	0.547 ± 0.220	0.691 ± 0.129	0.197	4 June 2004
Nutlets/spikelet stem	0.032 ± 0.018	0.046 ± 0.013	0.532	4 June 2004

Data represent treatment since early May 2004.

## Data Availability

The data presented in this study are available in the article.

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
