# Peer review of "UV-B Radiation Effects on the Alpine Plant Kobresia humilis in a Qinghai-Tibet Alpine Meadow"

_plants, 2022, doi:10.3390/plants11223102_

Round 1

Reviewer 1 Report

Manuscript Review

UV-B radiation effects on the alpine plant Kobresia humilis in a Qinghai-Tibet alpine

By Sheng-Bo Shi, Rui Shi, Tian-Cai Li and Dang-Wei Zhou

Submitted to Plants (Plants-1922129)

This paper presents the results of an investigation on the effects of UVB radiation on an alpine plant species. The manuscript requires appropriate revisions before it can be considered for publication.

L37-38. What about nitrous oxide?

L52. Sensitive to

L61. The hypothesis should be rewritten. How does UVB radiation adversely affect the plant?

L61-63. How do the local conditions increase the negative effects of UVB radiation?

L110. It needs to be rewritten.

L117. to determine

L119-124. Why did the authors store the plant samples for 21 days? Why not 24 hours?

L139. Were increased

L140. those of the ambient UV-B

L142. between the enhanced UV-B treatment and the control. Also, in the manuscript, “increased UVB” should be changed to “enhanced UVB”.

Table 1. Plant height should not be bold and underlined. The values must be followed by letters (a, b) to show significant differences between treatments.

Table 2. The replications should be explained clearly in the method.

L162. It should be rewritten.

Table 3. The values must be followed by letters (a, b) to show significant differences between treatments.

L171-172. It should be rewritten.

L176. What year?

Table 4. The values must be followed by letters (a, b) to show significant differences between treatments.

L182-184. It should be rewritten.

L188. Our results confirmed that K. humilis is a

L191-192. “some of replicated measurements” is not clear.

L198. “who found” but not “which found”

L269. Journal but not Journey

L282-283. The journal name should be checked.

L329-330. This reference should be written properly.

L357. DOI should be added to all references or deleted from here.

Author Response

We are very appreciated to three reviewers for kindly help on the manuscript. We have done the revision as follows:

Open Review 1

Submitted to Plants (Plants-1922129)

This paper presents the results of an investigation on the effects of UVB radiation on an alpine plant species. The manuscript requires appropriate revisions before it can be considered for publication.

Answering:

Thank you very much for your pertinent revision suggestions, we have made appropriate modification based on your detailed comments. We have added a necessary background introduction of the local environments, adjusted the words order, and clarified the research goals in hypothesis section.

According to your suggestion, we have changed expression of “increased UV-B” to “enhanced UV-B”, and revised other problems.

There were some questions that we cannot give modification in manuscript, however we gave an explanation as follows, hope it would be clear.

Considering the question of the extraction time of plant material in the method, that is, the plant samples are kept for 21 days. Firstly, the extracted leaves are needed to be kept in the dark to avoid photo-oxidation or decomposition; at the same time, the indoor low temperature can also avoid the possible volatilization of the solvent. Secondly, K. humilis is a perennial sedge (Cyperaceae), and its leaves are obviously hard and leathery; therefore, it is difficult to completely extract photosynthetic pigments in a shorter time in a standing condition, especially to be kept in low temperature environment. In the early trials, we found that it is generally needed to take about two weeks or 14 days to extract most of photosynthetic pigments and UV-absorbing compounds from leaves. Therefore, we chose 21 days with consideration of the work cycle; we can not only complete the field work, but also start the indoor new test work, and can ensure the complete leaching of the material.

About some of problems existed in 4 tables. We have modified some expressions, such as “increased UV-B”, and given a note to explain the replication of measurement. For the significant differences between treatments, we still used the expression of “P-value” to determine the difference. It seems clearer than other expression such as letters (e.g. a, b) or star (e.g. *, **). Because the exact value have been provided, the author could judge their significances not only just depend on the value of p value more or less than 0.05. Sometime, even p=0.1 would be also remarkable in biological experiments. Therefore we thought it seem more suitable to use p-value showing significant differences between treatments in our experiment.

Reviewer 2 Report

Review of Shi – UV-B radiation effects on the alpine plant Kobresia humilis in a Qinghai-Tibet alpine meadow

Remarks for the Authors

The paper by Shi on the effect of UV-B is carefully designed and well executed. It appears that Kobresia is among those species that show little physiological response to moderate increases in UV-B radiation.  The nature of the plant’s adaptation to its environment and especially its leaf structure may enable it to be insensitive to moderate dosages of UV-B radiation.  Although the results are negative, the work is a useful addition to our knowledge of the various impacts of UV-B.  However, there are some clarifications that are needed in the manuscript that should be addressed in a revision prior to acceptance for publication in Plants. 

What motivated the hypothesis that Kobresia humilis would be adversely affected by UV-B radiation?  Although the region has the hallmarks of environmental sensitivity, is there a particular reason this particular sedge was selected?  Was it based purely on its abundance in the meadow? 

The study was performed at a single increased dose of UV-B radiation of 15.8 kJ m-2 d-1.  Why was a single value selected?  The experimental dosage represents a moderate level of UV-B, but why was it selected?  Although it reflects a moderate decrease in ozone levels, is there a reason a 14% decrease in ozone was targeted?  Is Kobresia more sensitive at higher levels?  Has there been a graduate decrease in Kobresia in the region that correlates with any ozone depletion or increase in UV-B dose? 

Statistical analyses are provided for all measured parameters except those summarized in Table 2.  Given that the statistics show that no parameter except leaf width and thickness are different between ambient and augmented UV-B levels, are the biomass levels in Table 2 actually meaningfully different?  The values seem rather close, and well within the error of the measurements coupled with biological variation.

In model plants, the impact of UV-B in laboratories has been evaluated on new leaves, which tend to be more sensitive to UV-B radiation.  Was there any discrimination between new and established leaves in the study?

Author Response

We are very appreciated to three reviewers for kindly help on the manuscript. We have done the revision as follows:

Open Review 2

Remarks for the Authors

The paper by Shi on the effect of UV-B is carefully designed and well executed. It appears that Kobresia is among those species that show little physiological response to moderate increases in UV-B radiation.  The nature of the plant’s adaptation to its environment and especially its leaf structure may enable it to be insensitive to moderate dosages of UV-B radiation.  Although the results are negative, the work is a useful addition to our knowledge of the various impacts of UV-B.  However, there are some clarifications that are needed in the manuscript that should be addressed in a revision prior to acceptance for publication in Plants.

Answering:

Thank you very much for your kindly affirmation, your constructive comments on our manuscript are very useful. We have revised our manuscript based on your suggestions. Many thanks for your help.

What motivated the hypothesis that Kobresia humilis would be adversely affected by UV-B radiation?  Although the region has the hallmarks of environmental sensitivity, is there a particular reason this particular sedge was selected?  Was it based purely on its abundance in the meadow?

Answering:

We have added some necessary elaboration on the ecological condition of the plateau; it could lay out the scientific questions, and provided a better answering on our research motivation. At the same time, the introduction of meadow vegetation was added in the method section; the status of its dominant species and constructive species that was emphasized, so that the authors could understand the important of K. humilis we selected.

The study was performed at a single increased dose of UV-B radiation of 15.8 kJ m-2 d-1.  Why was a single value selected?  The experimental dosage represents a moderate level of UV-B, but why was it selected?  Although it reflects a moderate decrease in ozone levels, is there a reason a 14% decrease in ozone was targeted?  Is Kobresia more sensitive at higher levels?  Has there been a graduate decrease in Kobresia in the region that correlates with any ozone depletion or increase in UV-B dose?

Answering:

Field trials of supplemental UV-B radiation were designed in one incremental dose; this is mainly from limitation of simulation method.

First of all, the emission dose from lamps and the arrangement in the field are the main restriction for simulation trial or supplementary dose; if we want to set a serial of supplementary doses, the methods that can be used are nothing more than adjusting the power supply voltage and changing the arrangement style of the lamps or height to irradiated surface; but both are difficult to implement in the field site. Mostly, the supplementary light quality and the micro-environmental factors under the frame just near irradiated surface will be strongly changed, causing an increasing on experimental uncertainty. Therefore, like the most of simulation experiments that we have referenced, we also used one intensity dose in field trial. This design can provide square wavelength enhancement of UV-B intensity to follow nature solar irradiation changes based on the natural background solar ultraviolet radiation. Only two steps of UV-B dose are achieved,and there are no much influence of light quality and other environmental factors, such as shade.

Secondly, the simulated 14% ozone depletion is the level that our simulation device can be achieved under the background of local solar UV-B intensity; that is also in line with the general trend warned by the Montreal Protocol (1987), and the Beijing Declaration (1999). If the emission of ozone-depleting substances is not strictly restricted, the approximate level or range of depletion may be achieved soon. Fortunately, owing to the active efforts of various nation governments, there has not been a sustained decline in the past two decades. Therefore, one purpose of this manuscript is to remind or warn us that even small depletion of ozone can cause disturbance on the plants in the Qinghai-Tibet Plateau, and its impact on the ecosystem level is still difficult to predict.

Finally, since we did not conduct another higher-dose simulation study at the same time or later, it is difficult to predict whether K. humilis is more sensitive at higher doses condition. The results from other broad-leaf plants showed that there were differences among different species and leaves develop stages; and its changes at the ecosystem level or the trophic level may be more meaningful. In view of the fact that the continuous ozone reduction has been suppressed in a large range of earth sphere, and there is an improved trend under the framework of international conventions, the continuous simulation trial has not been carried out in our research region later. One of the purposes of this manuscript is to remind peaple to pay attention to the Earth environments where we live together, and avoid the occurrence of unfavorable and unpredictable events.

Statistical analyses are provided for all measured parameters except those summarized in Table 2.  Given that the statistics show that no parameter except leaf width and thickness are different between ambient and augmented UV-B levels, are the biomass levels in Table 2 actually meaningfully different? The values seem rather close, and well within the error of the measurements coupled with biological variation.

Answering:

Table 2 only showed the results from experiments in 2005. Because there were only three replicates in one year, no statistical difference analysis was performed. Before this study, we have also done similar measurements, and the data showed the trend was relatively consistent between the ambient and enhanced UV-B treatment and the specific data varied greatly in different years. This may be due to the difference in climatic conditions in different years, which in turn caused differences in accumulation of aboveground biomass in meadow vegetation.

In model plants, the impact of UV-B in laboratories has been evaluated on new leaves, which tend to be more sensitive to UV-B radiation.  Was there any discrimination between new and established leaves in the study?

Answering:

Yes, there existed differences in sensitivity of UV-B radiation at different growth stages or at different mature state leaves; the negative effect of UV-B appeared to be remarkable at the beginning of trial on May, while mature leaves appeared to be more resistant to supplemental UV-B radiation. This conclusion is mainly according to the study on photosynthetic physiology of broad-leaved plant Saussurea superba; we have done observation from different growth stages and whorled leaves for this species. In view of the low plant configure and small leaves construction, the physiological and biochemical analysis is difficult; we did not do much of work on K. humilis, including pigments related to antioxidant activity. Based on the status of dominant and constructor in the alpine meadow, we focused our purpose on differences in phenology and morphological configuration, and its accumulation of biomass; we want to determine whether there may be potential impacts on the ecosystem level.

Reviewer 3 Report

Although the manuscript looks interesting and is well written, I feel that the interest of the subject matter covered is not sufficient for Plants. The focus of the manuscript is very limited to a very specific area, making it very localistic.

On the other hand, in my opinion, more important parameters in UV defense should have been determined, such as pigments related to antioxidant activity. I also believe that a study of the temporal evolution of the parameters determined (as well as those related to antioxidant activity) could provide a more global and interesting view.

Author Response

We are very appreciated to three reviewers for kindly help on the manuscript. We have done the revision as follows:

Open Review 3

Comments and Suggestions for Authors

Although the manuscript looks interesting and is well written, I feel that the interest of the subject matter covered is not sufficient for Plants. The focus of the manuscript is very limited to a very specific area, making it very localistic.

Answering:

Your comments are very correct and helpful for our modifications; we sincerely felt thanks for your kindly help. The plant we focused in this manuscript is really very limitable; the research area is also not extended enough, and has obvious local characteristics. Frankly speaking, we have also studied other plant species at the same time, but the research directions were focused in different way; for example, the photosynthetic physiology in broad-leaved plant. We studied its responses to enhance UV-B radiation at different growth stages and whorled leaves. However, the K. humilis, as a dominant species and constructive species of alpine meadow, has small leaves and low plants structure; it is more difficult to collect and separate leaves in the field, so that it is difficult to meet the needs of physiological and biochemical analysis. With this in mind, we focused our research on differences in phenology and morphological configuration, and accumulation of aboveground biomass, in order to determine their potential impacts on the ecosystem.

The alpine meadow is a typical vegetation type on the Qinghai-Tibet Plateau, with extremely harsh environment. Any reaction of K. humilis will inevitably affect the structure and function of the alpine meadow ecosystem and it will also influence on the trophic level, causing a series of chain reactions. Therefore, based on its dominant and constructive position in alpine meadow, its importance cannot be ignored.

In addition, although our research area is limited to this unique geographical unit, the special status of the Qinghai-Tibet Plateau as the third pole in the world shows that it is extremely sensitive to global changes. As one aspect of global changes, the enhancement of solar ultraviolet radiation intensity near earth surface caused by ozone depletion that will also have a profound impact on alpine meadows, and its response has attracted much attention because of its vulnerability and widespread distribution.

On the other hand, in my opinion, more important parameters in UV defense should have been determined, such as pigments related to antioxidant activity. I also believe that a study of the temporal evolution of the parameters determined (as well as those related to antioxidant activity) could provide a more global and interesting view.

Answering:

The suggestions given by the reviewers are very helpful and beneficial for our further work. We have some works in this direction; unfortunately, in view of the plant characteristics, the work on photosynthetic physiology and antioxidant activity mainly focuses on companion species in meadow, such as Saussurea superba. Because K. humilis has small leaves and low plants height, it is difficult to collect leaves in the field site. In addition, the leaves are relatively hard and leathery, the pretreatment of the samples is difficult and the limited research results are not ideal. Based on the status of the dominant species and constructive species of K. humilis in the alpine meadow, we focused on its changes in phenology, morphological configuration, and final biomass differences. We believe that the research on this direction would be helpful in determining the potential impacts on the alpine meadow ecosystem.

Round 2

Reviewer 2 Report

(Comments for Authors)

The revised manuscript is improved and is acceptable for publication.

Reviewer 3 Report

Dear authors, 

you have made a remarkable effort to improve the manuscript, but in my opinion it is not of sufficient quality to be published in Plants.